# Accuracy of Eddy-Current and Radar Methods Used in Reinforcement Detection

**DOI:** 10.3390/ma12071168

**Published:** 2019-04-10

**Authors:** Łukasz Drobiec, Radosław Jasiński, Wojciech Mazur

**Affiliations:** Department of Building Structures, Silesian University of Technology; ul. Akademicka 5, 44-100 Gliwice, Poland; radoslaw.jasinski@polsl.pl (R.J.); wojciech.mazur@polsl.pl (W.M.)

**Keywords:** NDT methods, rebar location, eddy-current method, GPR method

## Abstract

This article presents results from non-destructive testing (NDT) that referred to the location and diameter or rebars in beam and slab members. The aim of paper was to demonstrate that the accuracy and deviations of the NDT methods could be higher than the allowable execution or standard deviations. Tests were conducted on autoclaved aerated concrete beam and nine specimens that were specially prepared from lightweight concrete. The most advanced instruments that were available on the market were used to perform tests. They included two electromagnetic scanners and one ground penetrating radar (GPR). The testing equipment was used to analyse how the rebar (cover) location affected the detection of their diameters and how their mutual spacing influenced the detected quantity of rebars. The considerations included the impact of rebar depth on cover measurements and the spread of obtained results. Tests indicated that the measurement error was clearly greater when the rebars were located at very low or high depths. It could lead to the improper interpretation of test results, and consequently to the incorrect estimation of the structure safety based on the design resistance analysis. Electromagnetic and radar devices were unreliable while detecting the reinforcement of small (8 and 10 mm) diameters at close spacing (up to 20 mm) and of large (20 mm) diameters at a close spacing and greater depths. Recommendations for practical applications were developed to facilitate the evaluation of a structure.

## 1. Introduction

Non-destructive testing that refers to the location and geometry of reinforcement in the structure is currently quite common while using different types of measuring equipment. The obtained test results are often used to make verifying calculations to determine the resistance of elements and decide whether they should not be further used or whether they should be reinforced. Rebar diameters and the precise location of reinforcement in the structure is required to calculate its resistance. For elements subjected to bending (slabs, beams) and compression and bending (columns), the reinforcement location affects internal forces, and thus the element strength. Even minor errors in detecting its location significantly change the result of resistance calculations and affect the structure safety. Moreover, non-destructive testing is employed at the acceptance of the performed facilities. It is particularly important that the accuracy of measurements is higher than the allowable execution deviations specified in standards. Therefore, tests on the structure reinforcement require information on the precision and limitations of the applied method and test equipment. All the above are useful in developing the programme for estimating the reliable building conditions. 

Manufacturers are improving their products, and the measurement accuracy is increasing. It is very important to specify the measurement accuracy and to do this before the beginning of evaluating works. Non-destructive testing (NDT) methods are particularly useful for testing a wide surface or many elements. Electromagnetic and radar methods are currently the most often used NDT methods in detecting reinforcement in the structure [1,2,3,4]. Both of the methods have their advantages and disadvantages. It can be assumed that advantages of the electromagnetic method are: the accuracy of measurements and the possibility for determining the diameter of reinforcement. Its disadvantages include a short range of measurements and some errors that are triggered by resolution, which are significant when the rebars are closely arranged (lap splices, bundles of rebars) [5,6]. The advantage of the radar method is the possibility of localising the reinforcement at great depths; and its disadvantages cover difficulties in measuring diameters measurements and some measurement errors of damp structures [7,8]. 

Despite the continuous development, electromagnetic and radar methods still have present limitations. Many papers in the literature [3,4,5,8] present advantages and possibilities for using electromagnetic and radar methods. Information regarding their limitations and measurement accuracy is rarely published. According to the definition [9], the accuracy is a compatibility level between the obtained result of a single measurement and the expected value that is related to the systematic and accidental errors. However, design standards and standards specifying structures reinforced with rebars permit some execution deviations. Thus, it is interesting whether the real accuracy of methods is within standard limits of execution deviations. When the measurement accuracy is higher than the allowable execution deviations, then the non-destructive testing methods should not be used.

The accuracy of non-destructive tests on the reinforcement geometry can be analysed when considering the measurements of rebar diameter and their position in perpendicular and parallel direction to the plane the tested element [3]. The measurement accuracy mainly depends on the dimension of concrete cover and the mutual spacing of adjacent rebars. This article describes results from testing the measurement accuracy of reinforcement diameter and geometry while using the most advanced instruments that are available on the market. The main aim of paper was to demonstrate whether the measurement accuracy and related measurement uncertainties could be higher than the allowable execution deviation. The tests were conducted on reinforced precast beam made of autoclaved aerated concrete beam and nine specimens specially prepared from lightweight concrete. The analysis involved the effect of the cover thickness and horizontal spacing of rebars on determining the number of rebars and measurement results for their diameters. Additionally, the aspect of measurement accuracy of concrete cover thickness and the effect of cover dimensions on the measurement accuracy were analysed. The obtained results were compared to deviations accepted by standards. The measurement accuracy of diameters and covers was analysed in accordance with the uncertainty of results when assuming suitable estimators of uncertainty. The comparison of tests results for the same elements that were obtained by different methods and the direct analysis of obtained results had a significant contribution to the development of methods of NDT.

## 2. Accuracy and Possibilities of Non-Destructive Testing

### 2.1. Electromagnetic Method

The accuracy of electromagnetic tests mainly depends on the depth, spacing, and arrangement of rebars toward the direction of scanning, the type of reinforcement, and the quality of concrete surface [10,11]. A type of spectral analysis and compensation procedures of systematic measuring errors also affect the accuracy of the tests.

The depth of rebars location has a significant impact on the accuracy of diameter measurements, and even on the accuracy of rebars location. The maximum depth, at which reinforcement can be detected, depends on the diameter of rebars. For typical rebar diameters of 6–25 mm, depending on the employed device, the biggest depth at which the reinforcement can be detected is 100–200 mm. The closer location of rebars to the scanned surface evidently ensures a greater accuracy of the measured and real diameter. The acceptable measurement accuracy (5%) of the reinforcement diameter is obtained when the rebars are located at a depth of ca. 60 mm. Unfortunately, electromagnetic methods are not effective in detecting the non-metallic reinforcement, which is becoming increasingly common [12,13]. 

Drobiec et al. [11] describe tests to detect plain and ribbed rebars of 12 diameter. The tests were done with four different devices. Variations of the measured thickness of the concrete cover were within 4 mm, and the measured reinforcement diameter differed by not more than one gradation. Sivasubramanian et al. [14] presented the analysis of measurement errors of the electromagnetic device in slabs having dimensions of 400 × 400 × 250 mm, which were made of high compressive strength concrete (average 71.3 N/mm^2^). One rebar was placed in each slab. Rebars of 12, 16, 20, 25, and 32 mm diameter and 70 and 85 mm covers were used (in total 10 of test models). Figure 1 illustrates the graph of the relative measuring error of the reinforcement diameter as a function of the concrete cover size. The acceptable error (<10%) was obtained for the cover thickness of up to ca. 40 mm. The measurement error of diameter of the order of 100% was obtained in tests regarding reinforcement at the depth of over 70 mm. It should be emphasized that, although the paper [15] is relatively recent, the used device is not a modern one. As no tests on models of concrete with lower strength were conducted, it is difficult to estimate the impact of the high compressive strength concrete. In typical reinforced concrete structures (beams, slabs, columns) with standard covers (*c* ~ 20–45 mm), electromagnetic devices that are available in practice can be used to perform tests with an accuracy of the measurement that is less than 10%. The gradation of rebar diameters causes difficulties in diameter differentiation.

### 2.2. Radar Method

In the radar method, the measuring range of the structure depth depends on concrete structure, a type of an antenna, and the frequencies of the excited impulse [15,16,17]. In typical devices, this range is up to 750 mm. The image contrast that was obtained from radar tests depends on the relative difference between dielectric constant values at the contact area between materials. There are no difficulties in interpreting the obtained image due to considerable differences in constant values for concrete and steel. Tests on reinforcement location that were conducted with radar method generate radargrams, that is, the record of all reflected signals registered during the passage of a measuring probe on the element surface. The reinforcement image on radargram is a distortion of course contour lines in the shape of hyperbola arms directed the radargram down.

The modern measurement systems perform automatic analyses of radargrams, convert radargrams taken close to each other in one image, and visualize reinforcement in the construction in a legible way for users. The additional software can be used to take a spatial image of the structure with the reinforcement. The radar devices do not provide direct information regarding reinforcement diameters. Perhaps measurements in a three-dimensional (3D) model can be a solution for this problem. However, some published articles present some mathematical correlations between the shape of a hyperbola that is shown on the radargram and the reinforcement diameter. Devices with the option of determining the reinforcement diameter are soon likely to appear on the market. However, the accuracy of the measurement method might be still a problem. Currently, the accuracy of the cover measurements with the radar method performed with the best devices is defined as ±5–10 mm.

The papers [18,19,20,21,22,23,24] describe that the shape of hyperbola illustrating the reinforcement on the radargram depends on the wave propagation velocity *υ* and passage time *t*_0_ of measuring the probe above the tested reinforcement. The time of the measuring probe passage is the time passing from the moment when the device records the reinforcement for the first time until the moment when it stops to record it. The velocity of the wave propagation affected the hyperbola curve, whereas the time of the probe passage affected the range of its arms. Knowing the parameter values a and b of the hyperbola b (Figure 2), their relation with the reinforcement diameter *ϕ*, wave propagation velocity *υ*, and time of probe passage *t*_0_, on the tested reinforcement can be expressed, as following:(1)$$a=t_{0}+\frac{\phi }{\upsilon },$$
(2)$$b=\frac{\upsilon }{2}\left(t_{0}+\frac{\phi }{\upsilon }\right)$$

The wave propagation velocity *υ* and passage time *t*_0_ of the measuring probe were recorded by a measuring device; and, parameters a and b were read from the radargram. Therefore, there were no obstacles to determine the reinforcement diameter *ϕ* from Equations (5) and (6). The paper [25] confirmed the possible use of that method for defining the reinforcement, as under laboratory conditions, the obtained measuring error of diameters of steel pipes and cables was in the order of 1.7–5.3%. 

Another way of defining the diameter was suggested in the paper [26]. It was assumed that the impulse distribution angle from the transmitting antenna was 90° (Figure 3). Such an assumption was useful in determining the rebar diameter without taking into account the wave propagation velocity *υ* and passage time *t*_0_ of the measuring probe above the tested reinforcement. The reinforcement diameter *ϕ* can be determined according to:(3)$$\phi =2\sqrt{2}L-2X-2Y$$
where: L, X, Y—geometric data according to Figure 3.

### 2.3. Combination of Electromagnetic and Radar Methods

Wiwatrojanagul et al. [27] and Wiwatrojanagul et al. [28] suggested connecting the advantages of the electromagnetic and radar methods. Tests were conducted on specimens that were placed in a pen and in concrete. Rebar diameters were tested in rebar laboratories. The electromagnetic method was found to not give correct test results, because the reinforcement was placed too close. On the basis of conducted tests, the empirical equations were developed to estimate the diameter of contact rebars in the rebar laboratory. The accuracy of the determined diameters was obtained at the level of 2.35%. Empirical equations were found to have some application limitations regarding the tested reinforcement and concrete. 

To sum up, the advantage of devices working according to the electromagnetic method was their high accuracy at small depths of measurements. A small range of diameter measurements, which was limited to about 6 cm, was their disadvantage Radar devices had much bigger range (even up to 75 cm). However, devices that are available on market do not offer the possibility of diameter measurements. In the world, some methods for diameter determining on the basis of radargram measurements obtained from the radar tests are being developed. Radar devices with the option of diameter measuring are likely to appear soon on the market.

## 3. Tested Specimens

### 3.1. Autoclaved Areated Concrete (AAC) Precast Lintels

To verify the accuracy of NDT methods, some tests were performed on widely used lintels that were made of AAC with the width of *b* = 180 mm, height *h* = 240 mm and the total length of *L* = 2000 mm. The strength of AAC elements was *f* = 4 N/mm^2^. Detailed results from the material tests on lintels can be found in the papers [29,30,31,32]. The lintel reinforcement was made of steel with a yield strength of 500 N/mm^2^ (B class according to EN 1992-1-1:2008 [33]). Longitudinal rebars had a diameter of 8 mm (three rebars down and two rebars up). Longitudinal reinforcement in the form of open stirrups that were made of rebars having a diameter of 4.5 mm. Stirrups were placed along the whole element at a constant spacing of 150 mm (Figure 4). The longitudinal reinforcement and stirrups were welded and then covered with corrosion-resistant protective coating made of resin. 

Two beams were selected for tests. To ensure the precise location of the reinforcement, both ends of tested beams were broken down and the reinforcement was measured, as in Figure 5. Tiny openings (diameter of 2 mm) were made in 11 places on lateral surfaces and at eight bottom points (between stirrups) to measure the real reinforcement cover—*c*_obs_ with the accuracy of Δc_obs_ = ± 0,1 mm. As the location of both beams was the same, tests were only conducted on one of them.

### 3.2. Lightweight Concrete Specimens

The impact of bar diameter and spacing on measurement accuracy was tested on nine models (Figure 6) that were prepared from lightweight concrete with density of 0.9 kN/m^3^ and strength of 10.2 N/mm^2^ after seven days, and of 18.1 N/mm^2^ after 28 days. Each model had three rebars that were arranged to ensure the distance between rebars equal to 20, 30, and 40 mm (dimension “a” in Figure 6). The diameter of used rebars was ø = 10; 16; and, 20 mm. The dimensions of models in a plain view were 240 × 440 mm and thickness of 40, 60, and 80 mm. Different thickness was to ensure the correct (two-sided) cover of rebars with concrete assuming that rebar distance from one of the surface was 20 mm. It is the minimum thickness that is accepted by the standard EN 1996-1-1 [33] for the exposure class XC1 (indoor) and the structural class S3. The wooden elements were precisely cut to provide the cover thickness of 20 mm. Reinforcement was laid using wooden spacers of relevant thicknesses (Figure 7a), and then stabilized in washers with screws (Figure 7b). Concrete was laid, mechanically compacted, and properly cured. Figure 8 shows models before and after concreting. Specimens symbols contained a letter S, diameter *d*, and space between rebars *a*. For example, S-15-30 identified a specimen that was reinforced with three rebars having a diameter of 16 mm and space between rebars that is equal to 30 mm.

## 4. Applied Test Equipment 

The tests were conducted using two electromagnetic scanners: PS 200 (manufacturer Hilti Corp., Schaan, Lichtenstein), Profometer 630 AI (manufacturer Proceq AG, Schwerzenbach, Switzerland) and one GPR device –GPR Live (manufacturer Proceq AG, Schwerzenbach, Switzerland). Figure 9 shows the testing apparatus. A scanning transducer of the electromagnetic device 1 was equipped with one circumferential transmitting coil and seven pairs of receiving coils. The receiving coils induce current with microammeters, and the received signal is then processed and analysed. The electromagnetic device no. 2 is similar in principle. The radar equipment 3 was equipped with antennas to perform tests, while using a signal with a variable frequency within a range of 0.2–4.0 GHz. Frequency was changed progressively in an automatic way during tests, and the max. acquisition time was 20 ns.

The measuring accuracy of each device was the same, Δc_obs_ = ±1.0 mm (according to information provided by device manufacturers). 

## 5. Testing Locations

### 5.1. AAC Precast Lintels

The tests were performed with each device by scanning the same points on the lateral and bottom surface of beams. Stirrup rebars were detected on the lateral surface, whereas the main reinforcement was tested from the bottom. During tests, line and area scans were conducted. Thus, the test consisted in moving the transducer along lines or areas that were routed on the surface of the tested element (Figure 10). Figure 11a illustrates scanned areas for stirrup testing and Figure 11b presents scanned areas for testing the main reinforcement. Figure 11c shows the location of a line scan and the point of measuring stirrups, whereas Figure 11d shows the location of a line scan for the main reinforcement.

### 5.2. Lightweight Concrete Specimens

Tests on lightweight concrete specimens, just as tests on lintels, consisted in scanning along lines or on surfaces that were routed on a given element using the transducer. The lightweight concrete specimens were tested using the electromagnetic Equipment (1) over the area of 150 × 300 mm illustrated in Figure 12a. The electromagnetic Equipment (2) was used to perform line scans at the mid-length of the specimen (Figure 12b), while area scans were conducted with the radar equipment over the area of 200 × 300 mm, as shown in Figure 12c. At the first stage, scans were conducted on the element surface where the concrete thickness was 20 mm, and then the element was turned to conduct further scans. Concrete cover was 10, 24, and 40 mm for rebars having a diameter of 10, 16, and 20 mm. Figure 13 shows the types of used equipment.

## 6. Test Results

### 6.1. AAC Precast Lintels (AAC)

#### 6.1.1. Stirrups Tests

The stirrups were tested at the beam side as line and area scans. Line scans could be only used to determine the reinforcement location (spacing and concrete cover). Figure 14 illustrates the comparison of exemplary results from line scans that were taken with electromagnetic devices and the radar device (radargram). A great conformity was found between the test results and real measurements of rebar location. Figure 15 presents examples of area scans taken with the electromagnetic device 1 and the radar device 3. The area scans conducted with two methods very clearly showed both stirrup reinforcement and longitudinal reinforcement.

#### 6.1.2. Tests on Longitudinal Rebars

The aim of testing the main reinforcement was to determine the cover size, the number, and diameters of rebars in the main reinforcement (diameters were determined only in case of electromagnetic scans). Figure 16 shows the comparison of results from electromagnetic tests. The area scan (Figure 16a) was taken with the electromagnetic device 1 and the line scan (Figure 10b) was taken with the electromagnetic device 2. Both of the devices did not detect the correct number of rebars. Device 1 detected two rebars instead of three, but it measured the diameter quite correctly. However, device 2 detected only one rebar and the measured diameter was equal to 18 mm. Figure 16 shows the results from tests performed with the GPR device. Area and line scans were taken. The line scan is presented as a radargram (Figure 17a) and the area scan as a map (Figure 17b). Only one longitudinal rebar was detected during both tests while using the radar method.

To sum it up, the test results for the longitudinal reinforcement, in which the rebars were closer to each other (an axial spacing of rebars was 30 mm, as shown Figure 4) were poorer than in the case of tests that were performed on stirrup reinforcement as described in point 6.1.1. None of devices used during various testing methods, correctly detected the number of longitudinal rebars. The most satisfactory result was observed for tests that were performed with the electromagnetic instrument 1.

### 6.2. Lightweight Concrete Specimens (Tests with Concret Cover 20 mm)

#### 6.2.1. Tests Using a 20 mm Cover 

Tests on lintel beams indicated significant errors in measuring the quantity of reinforcement and the number of rebars in the main reinforcement. Thus, additional tests were performed on lightweight models that were prepared for this purpose. The aim of those tests was to capture the relationship between rebar diameters and spacing and the accuracy of measurements. Figure 18 shows the results from tests conducted with the electromagnetic equipment (1). They contain the measured number and diameter ø_obs_ of rebars and average cover *c_obs_*.

As in case of lintel beams, the tests failed to detect the correct number of rebars spaced at *a_nom_* = 20 mm for specimens that were reinforced with rebars having a diameter ø_nom_ = 10 mm. In the specimen with rebars at the similar spacing, but reinforced with rebars ø_nom_ = 16 mm, three rebars could be noticed, but only two of them were detected with the equipment, which also overestimated the diameter of 30 mm. When rebars with a 20 mm diameter were used, the device identified as many as three rebars, even at the minimum spacing; however, the indicated diameter was twice smaller (10 mm) than the real value. At a larger spacing *a_nom_* = 30 and 40 mm, the electromagnetic equipment (1) detected the correct diameter of rebars, but some inaccuracies were found while measuring the covers.

Figure 19 shows the results from tests conducted with the electromagnetic equipment (2). As above, the results include the number of rebars, their measured diameters *d*, and the average size of tested concrete cover *c*. For the smallest diameter and spacing, the equipment detected only one rebar, just as in case of tests on precast beam. The equipment found three rebars in specimens that were reinforced with rebars of 16 and 20 mm diameter, spaced at more than 20 mm. Some deviations were observed in measured diameters and covers. Contrary to the electromagnetic equipment (1), the equipment (2) provided more accurate diameters with less precisely measured thickness of concrete cover. Differences in cover measurements were even in the order of up to 5 mm.

Figure 20 shows results from tests that were conducted with the electromagnetic equipment (3). The results include the number and average size of tested concrete cover *c_obs_*. The correct number of rebars was detected in all of the tested specimens. The measured value of covers also showed high compliance with the true value (20 mm) The radar equipment could not directly measure diameters, which was its drawback. Therefore, there are no rebar diameters in Figure 20.

#### 6.2.2. Tests Using Covers of Variable Thickness

Figure 21 shows the results from tests that were conducted with the electromagnetic equipment (1). The obtained results were comparable to tests using the cover with constant thickness (Figure 18). However, some differences were noticed. All of the rebars were detected in specimens with rebars of 10 mm diameter and concrete thickness reduced by 50%. However, at a spacing of 20 mm (Figure 21a) and 30 mm (Figure 21b), the rebar diameter was identified with a serious error of the order of 40% and 20%, respectively. Correct results were obtained for rebars at a clear spacing of 40 mm, just as for twice as big cover.

Tests on specimens with rebar of 16 mm diameter and the cover of 24 mm produced results that were comparable to those for the same specimens, but on the other side (at cover thickness of 20 mm). For rebars at a closer spacing, the device wrongly identified two rebars having a diameter of 30 mm (Figure 21d). It overmeasured the thickness of concrete cover by more than 30%. For rebars at a moderate spacing, the measurement errors were comparable to those from testing the concrete cover of 20 mm thickness. In the specimens with rebars at the largest spacing, the measured diameters of rebars were the same.

A doubled increase in cover thickness in specimens with rebars of 20 mm caused a more serious errors event at the closest spacing of rebars (Figure 21g) because their number was incorrectly identified. Higher measurement deviations were found in two other specimens when compared to tests using the cover with twice as small thickness.

Figure 22 shows the results from tests that were conducted with the electromagnetic equipment (2). Similarly as for the electromagnetic device (1), some differences were noticed in test results when compared to tests using the concrete cover of 20 mm thickness. For rebars with a diameter of 10 mm at the closest spacing (Figure 22a), their correct number was not identified. At a greater spacing, the equipment identified the correct number but overmeasured the rebar diameters. The results for specimens with rebars having a diameter of 16 mm were similar to those from testing the other side of the element at the concrete cover of 20 mm. Tests on specimens with rebars having a diameter of 20 mm indicated more significant inaccuracies in measurements in comparison to tests on rebars having a diameter of 10 mm. The device failed to identify the correct number of rebars at spacing of 20 mm (Figure 22g), whereas their correct number was detected when twice the concrete thickness was used. The test results were more satisfactory at larger spacing of rebars. The diameters were slightly undermeasured and the thickness of concrete cover was correct.

Figure 23 shows results from tests that were conducted with the radar equipment (3). The number of rebars was wrongly identified only in the case of specimens with rebars having a diameter of 10 mm and 20 mm, and at the closest spacing. However, measurements of concrete cover were quite accurate.

## 7. Discussion of Results 

### 7.1. AAC Precast Lintels

#### 7.1.1. Stirrups Tests

Direct measurements were taken to analyse the accuracy of conducted tests. The thickness of reinforcement cover in holes made with a 2 mm drill was subjected to direct measurements at the tested locations (Figure 7). Results from these measurements are presented in column 2 of Table 1.

Results from non-destructive testing of 11 inner stirrups of the beam are presented in Table 1 (three last columns of the Table), along with estimators of uncertainty according to JCGM 100:2008 [34]. The uncertainty of calibration was assumed to be equal to the accuracy of particular devices, and the uncertainty of an experimenter and the random uncertainty were neglected. The real average cover of stirrups was *c*_obs_ = 28.3 mm, with a standard deviation of 0.87 mm and variation coefficient of 3%. The estimated average reinforcement cover with the standard uncertainty was 3 ± 0.3 mm (the confidence level 68.3%), and, while taking into account the maximum uncertainty at the confidence level of 99.7%, the cover was equal to 28.3 ± 0.9 mm. The reinforcement nominal cover was *c*_nom_ = 25 mm, and the execution deviation allowed by the standard EN 13670 [35] was Δ_minus_ = 10 mm. The result obtained from direct measurements was within the determined limits. When considering other methods, the obtained average cover was 27 mm, 28.6 mm, and 27 mm, while the standard and maximum deviations did not exceed ±0.3 mm and ±1.0 mm. The obtained minimum covers measured with every NDT methods were bigger than the determined minimum cover while taking into account the dimension deviation. The reinforcement diameter was only determined with the electromagnetic scanner 1. The obtained result was equal to 6 mm, with the real diameter of 4.5 mm. However, the device manufacturer declared the detection of rebars with a diameter from 6 mm.

The diameters of stirrups were calculated on the basis of the obtained radargram and while using the Equations (1)–(3). Moving time *t_o_* was determined with the device, and other quantities that are required for equations were read from the radargram. For Equations (1) and (2), the calculated stirrup diameter was 5.6 mm, and in case of the Equation (3), the calculated diameter was 5.9 mm. Thus, the determined diameter was greater by 33% and 41%, respectively, than the real diameter.

#### 7.1.2. Tests on Longitudinal Rebars

Similarly, as in case of testing stirrups, rebar cover was directly measured and column 2 of Table 2 shows the results. Measurement results are presented in Table 2 (three last columns of the Table) along with estimators of uncertainty according to JCGM 100:2008 [34]. The same assumptions were made as for the uncertainty analysis of stirrup location. There were some problems with the comparison of measurement results with the real results, because each device detected fewer rebars than the real number. Finally, it was decided to compare the test results of the electromagnetic device 1 (Figure 10a) to real measurements made on extreme rebars in holes that were drilled in places where electromagnetic measurements were taken (eight measuring points). In case of devices 2 and 3 devices that detected only one rebar, values from holes were compared to the value measured for one rebar. Thus, the same values are presented for measuring points 1 and 5, 2 and 6, 3 and 7, 4, and 8 (Figure 10a) in Table 2. The real average cover of longitudinal rebars was *c*_obs_ = 29.2 mm, with the standard deviation of 0.7 mm and the variation coefficient of 2.4 mm. The estimated average reinforcement cover with the standard uncertainty was 29.2 ± 0.3 mm (the confidence level 68.3%), and taking into account the maximum uncertainty at the confidence level of 99.7%, the cover was equal to 29.2 ± 0.8 mm. 

The nominal cover of reinforcement was *c*_nom_ = 25 mm, and the execution deviation allowed by the standard EN 13670:2011 [35] was Δ_minus_ = 10 mm. Thus, the obtained result from the direct tests was within the determined limits. For other methods, medium-sized covers equal to 34 mm, 27.5 mm, and 31 mm were obtained. Standard deviations mostly did not exceed ± 1 mm and at a maximum ± 4 mm. The obtained minimum covers were bigger than minimum values while taking into account executive deviations. 

In case of main reinforcement testing (beams at the bottom), the obtained deviations were considerably greater than in the case of stirrup tests (beams on the lateral surface). It should be emphasized that every device that was used in the tests did not detect the real number of rebars in the main reinforcement. Consequently, the result was burdened with a serious error resulting from the wrong reading of rebars number by research devices.

### 7.2. Lightweight Concrete Specimens

Table 3 presents the results for lightweight concrete models that were tested on the side where the cover thickness was constant and equal to 20 mm, whereas Table 4 shows the results for the variable thickness of the cover. A shaded field shows measured values that were the same as the nominal values in models. The analysis of values in the table indicates that the number of rebars having a diameter of 10, 16, and 20 mm could be identified at a spacing greater than 20 mm and cover thickness greater than 10 mm while using the electromagnetic equipment (1) and (2). Measured values closest to nominal values of diameters were observed at a rebar spacing of 30 mm. The equipment showed similar diameters, but the false number of rebars at a closer spacing. The average ratio *ø_obs_*/*ø_nom_* in accordance with electromagnetic methods (1) and (2), was 1.1 and 0.93, respectively, for the cover of 20 mm, and 1.08 and 1.02 for the cover of variable thickness. If the accuracy of indications of the electromagnetic equipment of the order of ca. ±1.0 mm was taken into account during the analysis of measurements, then the results from measuring diameters with these methods could be considered as reliable at a spacing larger than 20 mm and for diameters that are greater than 10 mm. Rebar diameters could not be read in the case of using the radar equipment (3). Therefore, the diameters were calculated in a similar way to precast lintel beam, while using Equations (1)–(3). More satisfactory compliance with nominal values was achieved using Equations (1) and (2). Table 3 shows those values. The measured diameters were greater by ca. 15% than nominal values.

The rebar covers could be measured while using all of those methods. At the nominal cover with thickness of 10 mm, average values of 11.3, 9.07, and 11.3 could be read from the equipment (1), (2), and (3). For specimens with a 20 mm cover, the average measured values of cover thickness were 21.6, 17.3, and 21.1 mm, depending on the used device. for slightly greater cover (24 mm), the identified average values were 27.7, 20.4, and 23.3 mm; whereas, a double increase in concrete thickness to 40 mm resulted in average values that were equal to 43.3, 31.9, and 39.7 mm. Thus, the radar equipment was found to measure concrete thickness with the greatest precision.

In all measurements of covers, the difference from the nominal value was ±6,5 mm, that is, within the limits of allowable execution deviations Δ_minus_ = 10 mm specified by the standard EN 13670:2011 [35]. 

Tests that were conducted on lightweight concrete specimens confirmed the results obtained from tests on precast lintel beam. The greatest inaccuracies of measurements were observed for rebars with lower diameters (≤10 mm) at close spacing (≤20 mm), and with great diameters (20 mm) at close spacing (20 mm) and great depth (40 mm). For greater diameters (≥16 mm), even close spacing did not affect the detection of the reinforcement, including the number of rebars, their diameter, and the thickness of concrete cover. The radar equipment provided a more precise size of concrete cover when compared to the electrical equipment. Rebar diameters that were calculated from the radargram were overcalculated by ca. 15 in comparison to nominal values. However, considerably better results in measuring diameters were noticed in the case of electromagnetic diameters.

## 8. Conclusions

The conducted tests demonstrated the occurrence of some limitations of two popular methods for detecting reinforcement location in the structure. The devices were able to correctly detect rebars, even those of small diameters, providing that they were located at an adequate spacing. Devices for scanning reinforcement had some problems with defining the correct quantity and diameter of rebars placed close to each other, at a distance shorter than 2–3 diameters.

The results obtained for covers measured with electromagnetic and radar devices did not significantly differ in terms of an average value. For the real cover, the differences did not exceed the acceptable level of few percent (<5%). There was no clear tendency to state that any of the methods artificially distorted indications, for instance, due to the simplification for measurement method validation. However, it should be emphasized that the minimal covers determined by all non-destructive techniques were bigger than the minimal cover specified with regard to size deviations (accepted by standard EN 13670:2011 [35]) and obtained by direct measurements. Thus, the tests on existing building can generate an error that is greater than the acceptable execution deviation. This can lead to wrong conclusions regarding the accuracy of a building. Measurements that were taken in the existing structure should be analysed with caution because the incorrect interpretation of measurement results for reinforcement location in the cross-section can cause a reduction in safety coefficient for steel *γ**_S, red_*_1_ from 1.15 to 1.1 (acc. to EC-2 [33]) and an unintentional reduction of safety level in particularly significant support zones of reinforced concrete structures and prestressed structures.

The tests indicated that a measurement error was significant, particularly for measuring structures that were reinforced with small diameter bars at a small spacing. This could lead to improper interpretation of test results and, consequently, to wrong calculation analyses for structures. A similar error could be observed in the tests on reinforcement of greater diameter, with rebars at a closer spacing when the concrete thickness was twice the rebar diameter or spacing.

To sum it up, the following recommendations for performing tests on reinforcement location can be considered as the practical application of the obtained results:a)while detecting the cover with thickness < 20 mm, errors in determining the number and diameter of rebars can be expected regardless of the used test equipment,b)while detecting the reinforcement at a depth > 20 mm, the employed equipment can correctly determine the number of rebars if the spacing is greater than 20 mm.c)NDT equipment is not suitable for detecting the reinforcement in support zones of beams and slabs, and in joints where reinforcement is the densest, andd)this equipment is perfect for detecting the longitudinal reinforcement in slabs, and stirrups in beams.

Studies aiming at connecting both methods, consisting in connecting scans that were taken with devices operating according to the electromagnetic and radar method seem to be reasonable. Hybrid devices are likely to generate accurate results, especially in terms of measuring the reinforcement diameters. 

## Figures and Tables

**Figure 1 materials-12-01168-f001:**
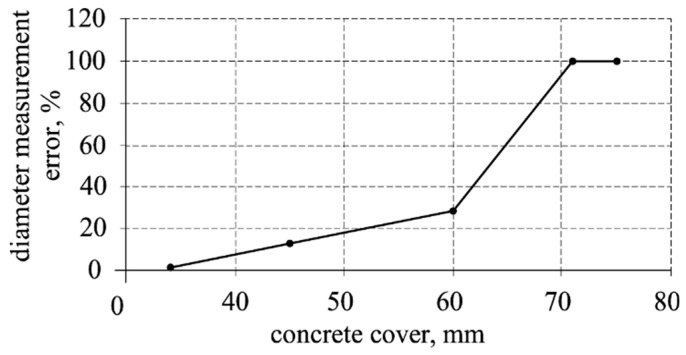
The error of the reinforcement diameter measured with an electromagnetic device, depending on the cover size according to [14].

**Figure 2 materials-12-01168-f002:**
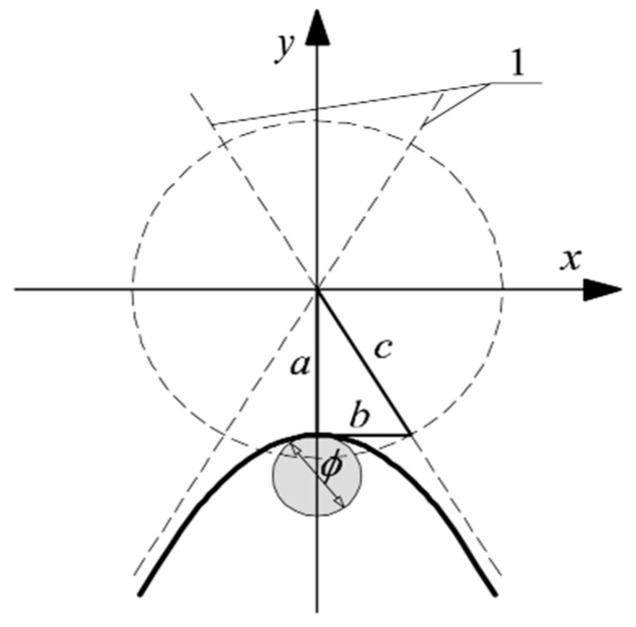
Assumptions to derive Equations (5) and (6), 1—hyperbola asymptotes.

**Figure 3 materials-12-01168-f003:**
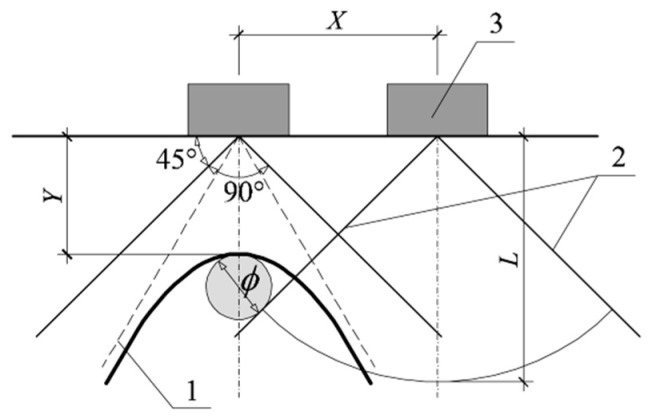
Assumptions to derive the Equation (7), 1—hyperbola asymptotes, 2—the angle of impulse distribution from the antenna, 3—a measuring probe.

**Figure 4 materials-12-01168-f004:**
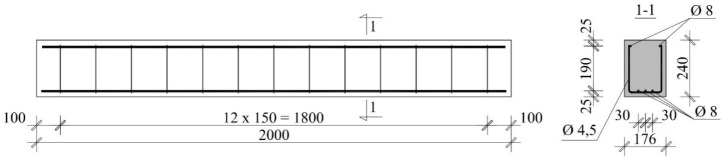
Dimensions and reinforcement of the tested lintel.

**Figure 5 materials-12-01168-f005:**
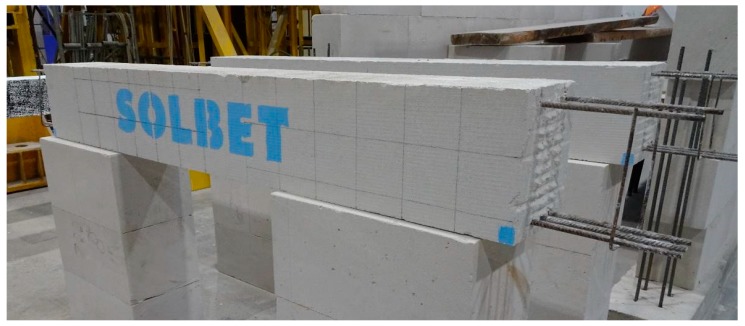
Tested elements with the reinforcement broken down at its ends.

**Figure 6 materials-12-01168-f006:**
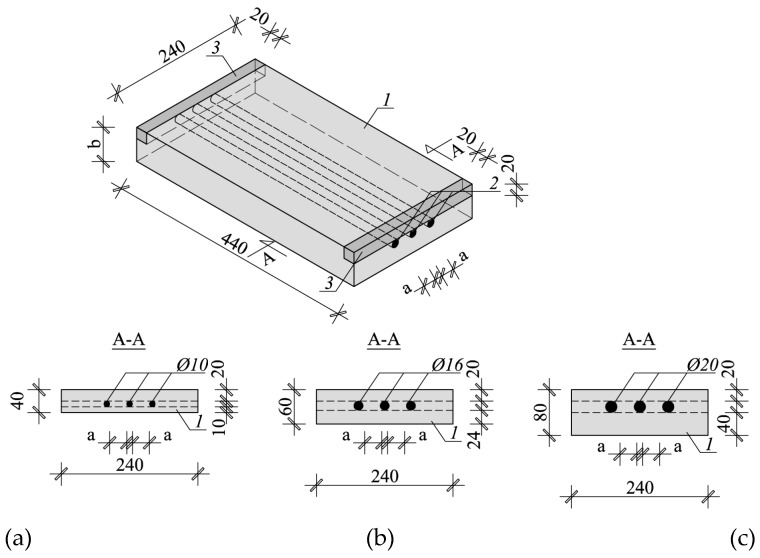
Lightweight concrete specimens: (**a**) specimen reinforced with rebars of 10 mm diameter, (**b**) specimen reinforced with rebars of 16 mm diameter, (**c**) specimen reinforced with rebars of 20 mm diameter, 1—lightweight concrete, 2—rebars, 3—wooden washer.

**Figure 7 materials-12-01168-f007:**
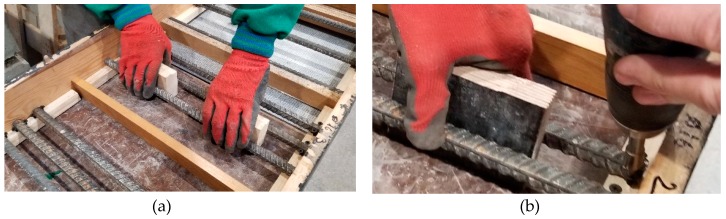
Stabilization of reinforcement in formwork: (**a**) distance resulting from wooden spacers, (**b**) rebar fastening to wooden washers with screws.

**Figure 8 materials-12-01168-f008:**
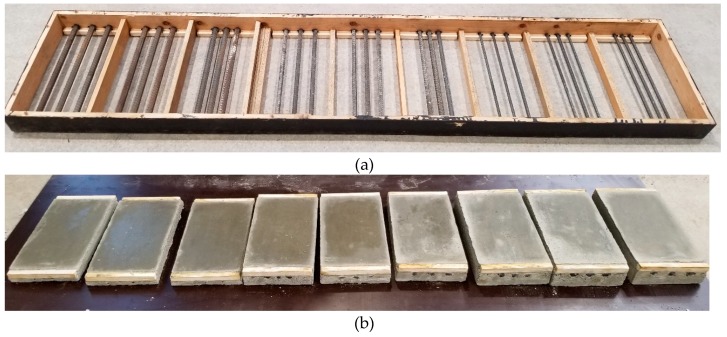
Specimen reinforcement in formwork (**a**) and specimens after concreting (**b**).

**Figure 9 materials-12-01168-f009:**
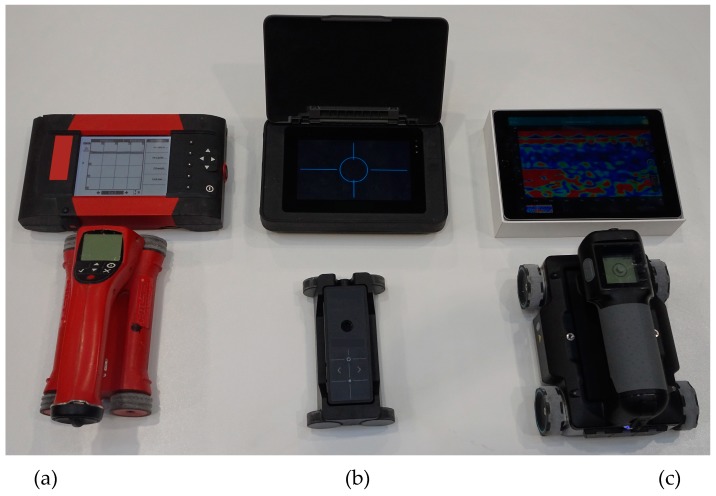
Devices used in tests, (**a**) electromagnetic device (1), (**b**) electromagnetic device (2), and (**c**) radar device (3).

**Figure 10 materials-12-01168-f010:**
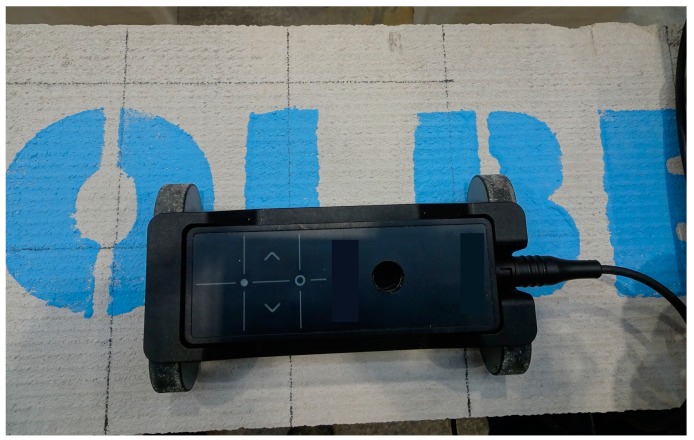
Transducer on the surface of the tested element.

**Figure 11 materials-12-01168-f011:**
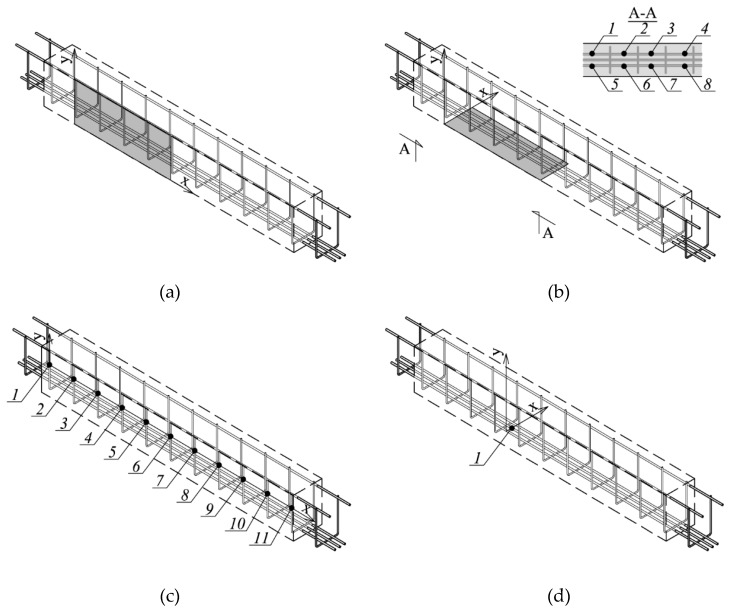
Places of lintel tests: (**a**) Stirrup tests—area scan (measurement in the grey-coloured field), (**b**) main reinforcement test—area scan (measurement in the grey-coloured field, measurement at points 1–8), (**c**) Stirrup tests—line scan (measurement at points 1–11), and (**d**) main reinforcement test—line scan (measurement at point 1).

**Figure 12 materials-12-01168-f012:**
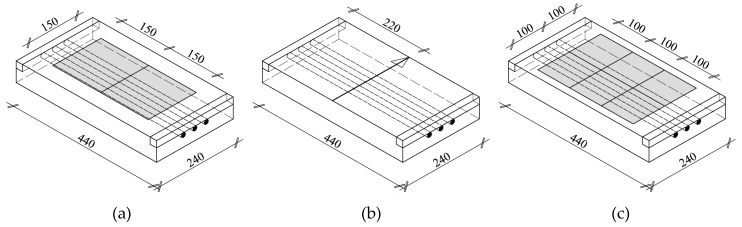
Location of tests performed with: (**a**) electromagnetic device (1), (**b**) electromagnetic device (2), (**c**) radar device (3).

**Figure 13 materials-12-01168-f013:**
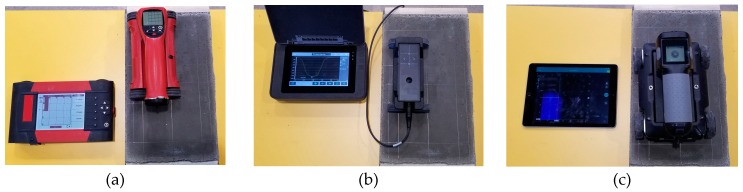
Testing specimens using: (**a**) electromagnetic device (1), (**b**) electromagnetic device (2), and (**c**) radar device (3).

**Figure 14 materials-12-01168-f014:**
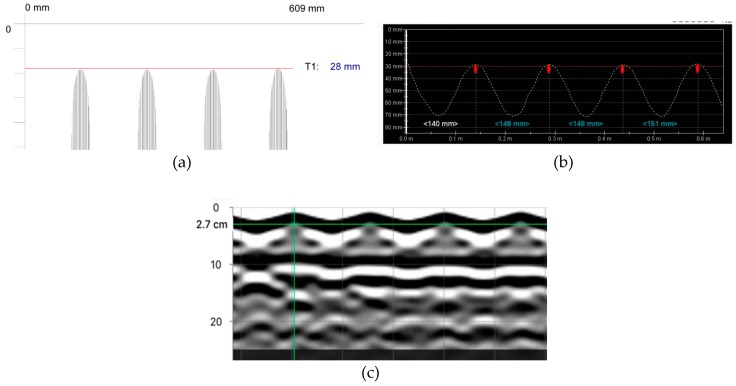
Line scans (measurement at points shown in Figure 10c) with devices: (**a**) electromagnetic 1, average cover of visible results—27.8 mm, (**b**) electromagnetic 2, average cover—28.6 mm, (**c**) radar 3, average cover 27 mm.

**Figure 15 materials-12-01168-f015:**
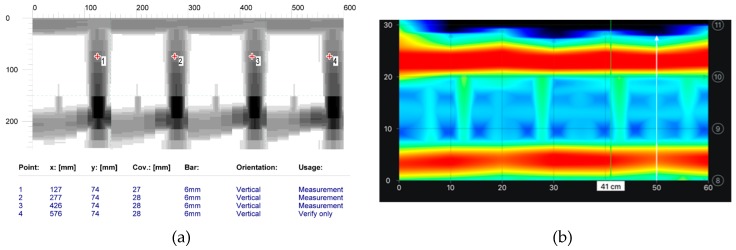
Area scans conducted with electromagnetic devices (measurement on the surface shown in Figure 10a), (**a**) electromagnetic 1, average cover of visible results– 27.8 mm, (**b**) radar 3,– 27 mm.

**Figure 16 materials-12-01168-f016:**
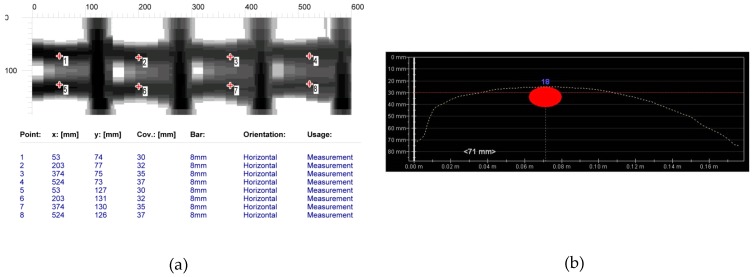
Scans taken with electromagnetic devices, (**a**) device 1, two rebars detected, average cover of visible results—33.6 mm, diameter 6-8 mm (measurement on the surface shown in Figure 10b), (**b**) device 2, one rebar detected, average cover of visible results—28 mm, diameter 18 mm (measurement at the point shown in Figure 11d).

**Figure 17 materials-12-01168-f017:**
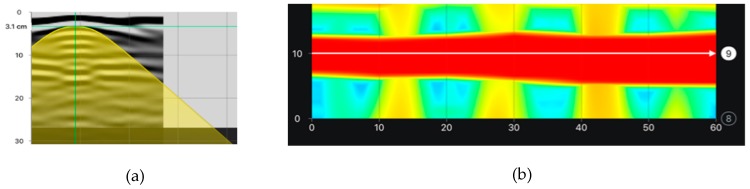
Scans taken with the radar (3), one rebar detected, (**a**) line scan (radargram—measurement at the point shown in Figure 10d), (**b**) area scan (measurement on the surface shown in Figure 10b).

**Figure 18 materials-12-01168-f018:**
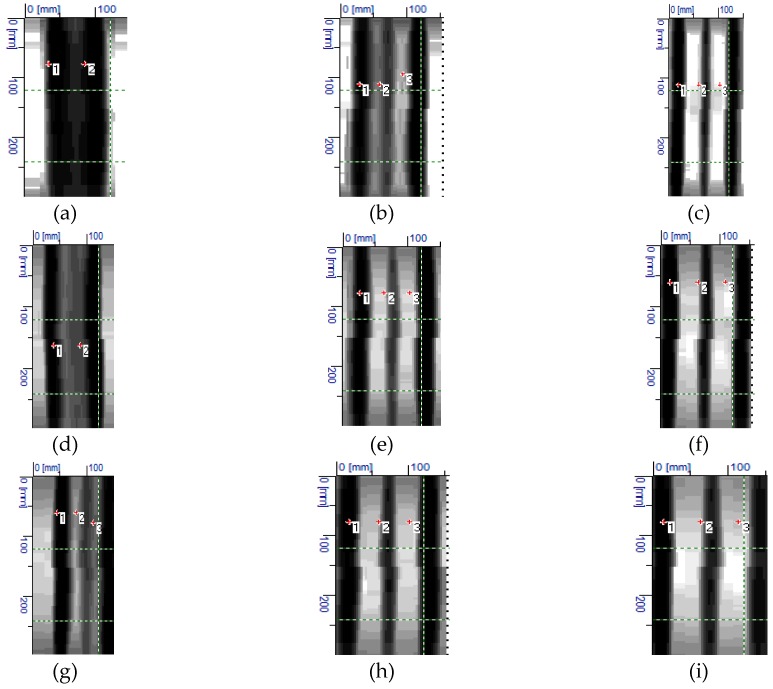
Results from tests conducted with the electromagnetic device (1): (**a**) S-10-20 (result: two rebars, ø_obs_ = 8 mm, *c*_obs_ = 25 mm), (**b**) S-10-30 (three rebars, ø_obs_ = 8 mm, *c*_obs_ = 20 mm), (**c**) S-10-40 (three rebars, ø_obs_ = 10 mm, *c*_obs_ = 21 mm), (**d**) S-16-20 (two rebars, ø_obs_ = 30 mm, *c*_obs_ = 25 mm), (**e**) S-16-30 (three rebars, ø_obs_ = 16 mm, *c*_obs_ = 18 mm), (**f**) S-16-40 (three rebars, ø_obs_ = 20 mm, *c*_obs_ = 20 mm), (**g**) S-20-20 (three rebars, ø_obs_ = 20 mm, *c*_obs_ = 25 mm), (**h**) S-20-30 (three rebars, ø_obs_ = 20 mm, *c*_obs_ = 18 mm), and (**i**) S-20-40 (three rebars, ø_obs_ = 20 mm, *c*_obs_ = 22 mm).

**Figure 19 materials-12-01168-f019:**
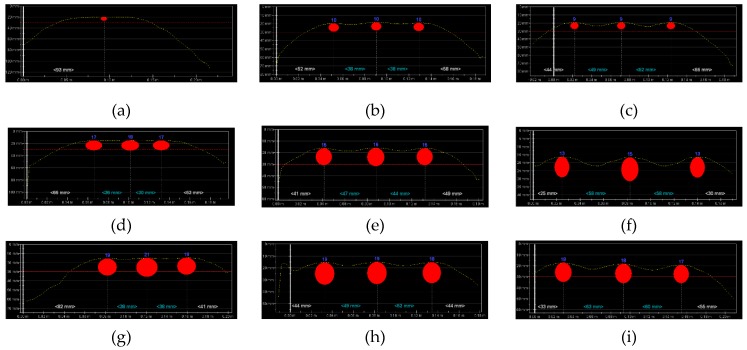
Results from tests performed with the electromagnetic device (2): (**a**) S-10-20 (result: one bar, ø_obs_ = 8 mm, *c*_obs_ = 20 mm), (**b**) S-10-30 (three rebars, ø_obs_ = 10 mm, *c*_obs_ = 19.3, mm), (**c**) S-10-40 (three rebars, ø_obs_ = 9 mm, *c*_obs_ = 19.3 mm), (**d**) S-16-20 (three rebars, ø_obs_ = 17 mm, *c*_obs_ = 15.5, mm), (**e**) S-16-30 (three rebars, ø_obs_ = 15 mm, *c*_obs_ = 16.4, mm), (**f**) S-16-40 (three rebars, ø_obs_ = 14 mm, *c*_obs_ = 16.5, mm), (**g**) S-20-20 (three rebars, ø_obs_ = 19 mm, *c*_obs_ = 15 mm), (**h**) S-20-30 (three rebars, ø_obs_ = 19 mm, *c*_obs_ = 15.4 mm), and (**i**) S-20-40 (three rebars, ø_obs_ = 18 mm, *c*_obs_ = 18.3 mm).

**Figure 20 materials-12-01168-f020:**
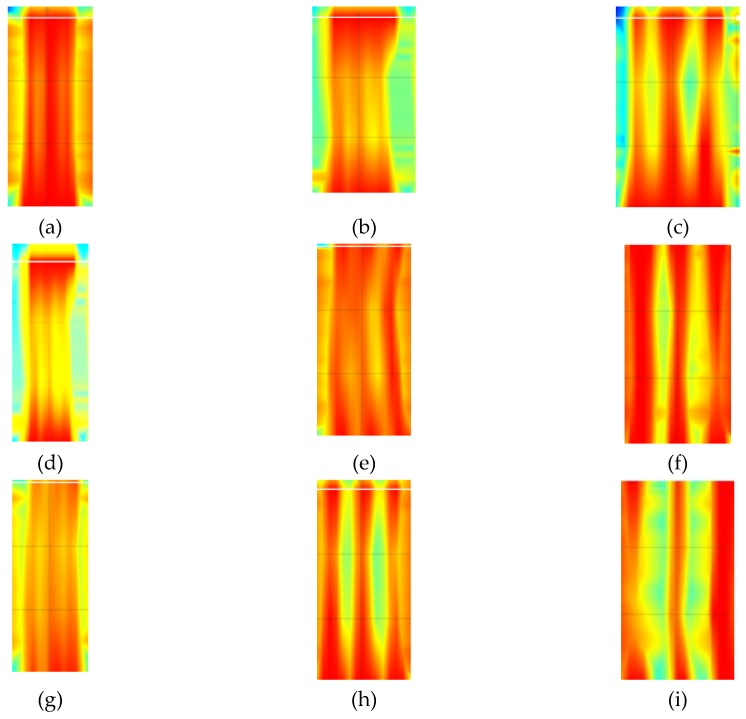
Results from tests performer with the radar equipment (3): (**a**) S-10-20 (result: three rebars, *c*_obs_ = 22 mm), (**b**) S-10-30 (three rebars, *c*_obs_ = 22 mm), (**c**) S-10-40 (three rebars, *c*_obs_ = 21 mm), (**d**) S-16-20 (three rebars, *c*_obs_ = 22 mm), (**e**) S-16-30 (three rebars, *c*_obs_ = 21 mm), (**f**) S-16-40 (three rebars, *c*_obs_ = 20 mm), (**g**) S-20-20 (three rebars, *c*_obs_ = 22 mm), (**h**) S-20-30 (three rebars, *c*_obs_ = 20 mm), (**i**) S-20-40 (three rebars, *c*_obs_ = 20 mm).

**Figure 21 materials-12-01168-f021:**
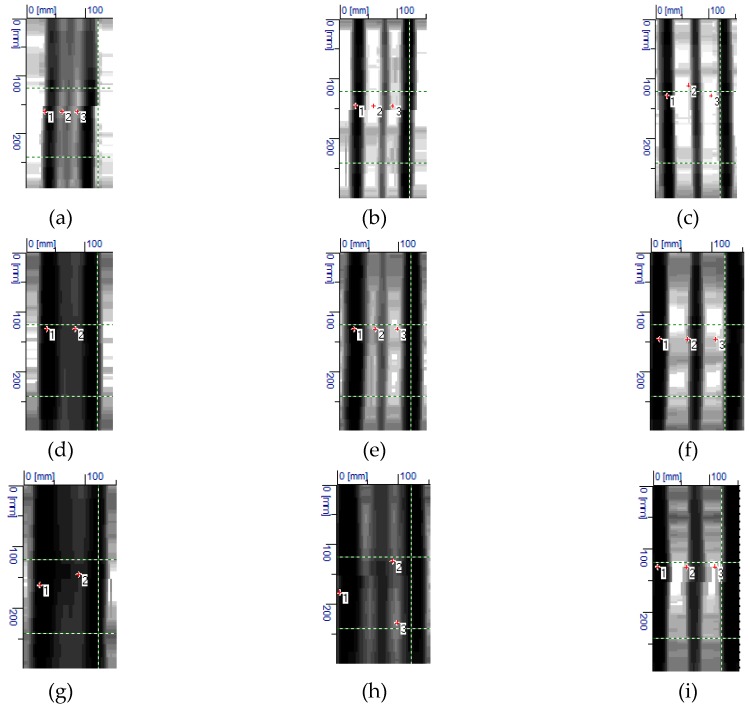
Results from tests conducted with the electromagnetic device (1): (**a**) S-10-20 (result: three rebars, ø_obs_ = 6 mm, *c*_obs_ = 11 mm), (**b**) S-10-30 (three rebars, ø_obs_ = 12 mm, *c*_obs_ = 12 mm), (**c**) S-10-40 (three rebars, ø_obs_ = 10 mm, *c*_obs_ = 11 mm), (**d**) S-16-20 (two rebars, ø_obs_ = 30 mm, *c*_obs_ = 32 mm), (**e**) S-16-30 (three rebars, ø_obs_ = 16 mm, *c*_obs_ = 27 mm), (**f**) S-16-40 (three rebars, ø_obs_ = 16 mm, *c*_obs_ = 24 mm), (**g**) S-20-20 (two rebars, ø_obs_ = 20 mm, *c*_obs_ = 48 mm), (**h**) S-20-30 (three rebars, ø_obs_ = 20 mm, *c*_obs_ = 47 mm), (**i**) S-20-40 (three rebars, ø_obs_ = 20 mm, *c*_obs_ = 35 mm).

**Figure 22 materials-12-01168-f022:**
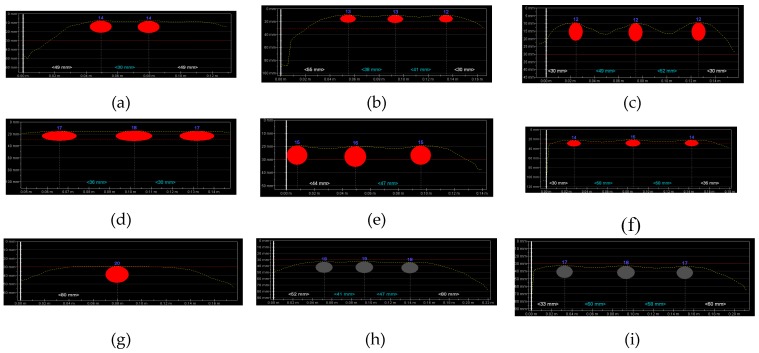
Results from tests performed with the electromagnetic device (2): (**a**) S-10-20 (result: two rebars, ø_obs_ = 14 mm, *c*_obs_ = 7.8 mm), (**b**) S-10-30 (three rebars, ø_obs_ = 13 mm, *c*_obs_ = 9.7 mm), (**c**) S-10-40 (three rebars, ø_obs_ = 12 mm, *c*_obs_ = 9.7 mm), (**d**) S-16-20 (three rebars, ø_obs_ = 17 mm, *c*_obs_ = 19.5, mm), (**e**) S-16-30 (three rebars, ø_obs_ = 16 mm, *c*_obs_ = 19.8 mm), (**f**) S-16-40 (three rebars, ø_obs_ = 14 mm, *c*_obs_ = 21.9 mm), (**g**) S-20-20 (one rebar, ø_obs_ = 20 mm, *c*_obs_ = 29 mm), (**h**) S-20-30 (three rebars, ø_ob_ = 18 mm, *c*_obs_ = 33.5 mm), (**i**) S-20-40 (three rebars, ø_obs_ = 17 mm, *c*_obs_ = 33.1 mm).

**Figure 23 materials-12-01168-f023:**
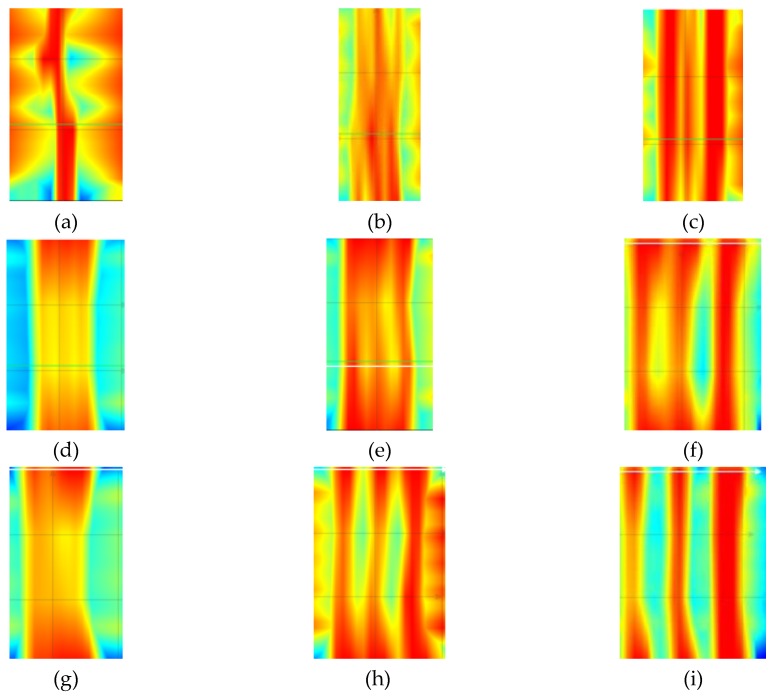
Results from tests performed with the radar equipment (3): (**a**) S-10-20 (result: one rebar, *c*_obs_ = 11 mm), (**b**) S-10-30 (three rebars, *c*_obs_ = 12 mm), (**c**) S-10-40 (three rebars, *c*_obs_ = 11 mm), (**d**) S-16-20 (three rebars, *c*_obs_ = 23 mm), (**e**) S-16-30 (three rebars, *c*_obs_ = 24 mm), (**f**) S-16-40 (three rebars, *c*_obs_ = 23 mm), (**g**) S-20-20 (two rebars, *c*_obs_ = 38 mm), (**h**) S-20-30 (three rebars, *c*_obs_ = 40 mm), (**i**) S-20-40 (three rebars, *c*_obs_ = 41 mm).

**Table 1 materials-12-01168-t001:** Measurement results for stirrup reinforcement using each method.

Measuring Point no.	Cover Measured with the Method, mm			
	Direct Method, *c*_obs_	Electromagnetic Device (1)	Electromagnetic Device (2)	Radar Device (3)
1	30.1	27	28.9	27
2	28.1	28	28.6	27
3	28.3	28	29.1	27
4	28.2	28	28.1	27
5	29.1	27	28.6	28
6	27.3	27	28.2	27
7	27.9	27	28.7	27
8	26.9	28	28.3	27
9	28.3	27	28.4	27
10	28.2	27	28.6	27
11	29.0	28	28.7	28
Average cover, *c*_obs_:	28.3	27	28.6	27
Nominal cover, *c*_nom_:	15 mm ≤ *c*_nom_ ≤ 25 mm			
Calibration uncertainties Δc_obs_	0.1	1.0	0.1	1.0
Standard deviation, *s*:	0.87	0.52	0.30	0.40
Standard deviation of the mean value $\bar{s}=\frac{s}{\sqrt{n}}$	0.3	0.2	0.1	0.1
Standard deviation of the calibration $S_{d}=\frac{\Delta c_{obs}}{\sqrt{3}}$	0.1	0.6	0.1	0.6
Standard uncertainty $S_{c}=\sqrt{S_{d}^{2}+{\bar{s}}^{2}}$	0.3	1	0.1	1
Result with standard uncertainty $c_{mv}\pm S_{c}$	28.3 ± 0.3	27 ± 1	28.6 ± 0.1	27 ± 1
Maximum uncertainty $\Delta c=\Delta c_{obs}+3\bar{s}$	0.9	1	0.4	1
Result with maximum uncertainty $c_{mv}\pm \Delta c$	28.3 ± 0.9	27 ± 1	28.6 ± 0.4	27 ± 1
Minimum cover *c*_min_	27.4	26	28.2	26

**Table 2 materials-12-01168-t002:** Measurement results for the main reinforcement using each method.

Measuring Point no.	Cover Measured with the Method, mm			
	Direct Method, *c*_obs_	Electromagnetic Device (1)	Electromagnetic Device	Radar Device (3)
1	29.1	30	26.2	31
2	28.8	32	27.3	30
3	28.7	35	28.1	31
4	29.2	37	28.4	32
5	29.1	30	26.2	31
6	28.9	32	27.3	30
7	29.2	35	28.1	31
8	30.9	37	28.4	32
Average cover, *c*_obs_:	**29.2**	**34**	**27.5**	**31**
Nominal cover, *c*_nom_:	15 mm ≤ *c*_nom_ ≤ 25 mm			
Calibration uncertainties Δc_obs_	0.1	1.0	0.1	1.0
Standard deviation, *s*:	0.70	2.9	0.9	0.8
Standard deviation of the mean value, $\bar{s}=\frac{s}{\sqrt{n}}$	0.2	1.0	0.3	0.3
Standard deviation of the calibration $S_{d}=\frac{\Delta c_{obs}}{\sqrt{3}}$	0.1	0.6	0.1	0.6
Standard uncertainty $S_{c}=\sqrt{S_{d}^{2}+{\bar{s}}^{2}}$	0.3	1	0.3	0.6
Result with standard uncertainty $c_{mv}\pm S_{c}$	**29.2 ± 0.3**	**34 ± 1**	**27.5 ± 0.3**	**31 ± 0.6**
Maximum uncertainty $\Delta c=\Delta c_{obs}+3\bar{s}$	0.8	4	1.1	2
Result with maximum uncertainty $c_{mv}\pm \Delta c$	**29.2 ± 0.8**	**34 ± 4**	**27.5 ± 1.1**	**31 ± 2**
Minimum cover *c*_min_	28.4	30	26.4	29

**Table 3 materials-12-01168-t003:** Test results for lightweight concrete specimens (at constant cover thickness of 20 mm).

Geometry of Models				Results from Tests Conducted with the Equipment (1), (2) or (3)								
Reinforcement Diameter (mm)	Number of Rebars *n*_nom_	Rebar Spacing *a*_nom_, mm	Reinforcement Cover *c*_nom_, mm	ø_obs_, mm			*n*_obs_, mm			*c*_obs_, mm		
				(1)	(2)	(3) *	(1)	(2)	(3)	(1)	(2)	(3)
10	3	20	20	8	8	11.9	2	1	3	25	20	22
		30		8	10	11.8	3	3	3	20	19.3	22
		40		10	9	11.5	3	3	3	21	19.3	21
16	3	20	20	30	17	18.8	2	3	3	25	15.5	22
		30		16	15	18.2	3	3	3	18	16.4	21
		40		20	14	17.8	3	3	3	20	16.5	20
20	3	20	20	20	19	23.3	3	3	3	25	15	22
		30		20	19	23.1	3	3	3	18	15.4	20
		40		20	18	22.8	3	3	3	22	18.3	20

(1)– electromagnetic device (Figure 13a), (2) – electromagnetic device (Figure 13b), (3) – radar device (Figure 13c), * – values obtained from Equations (1) and (2).

**Table 4 materials-12-01168-t004:** Test results for lightweight concrete specimens (at variable cover).

Geometry of Models				Results from Tests Conducted with the Equipment (1), (2) or (3)								
Reinforcement Diameter (mm)	Number of Rebars *n*_nom_	Rebar Spacing *a*_nom_, mm	Reinforcement Cover *c*_nom_, mm	ø_obs_, mm			*n*_obs_, mm			*c*_obs_, mm		
				(1)	(2)	(3) *	(1)	(2)	(3)	(1)	(2)	(3)
10	3	20	10	6	14	12.0	3	2	1	11	7.8	11
		30		12	13	11.9	3	3	3	12	9.7	12
		40		10	12	11.6	3	3	3	11	9.7	11
16	3	20	24	30	17	18.9	2	3	3	32	19.5	23
		30		16	16	18.8	3	3	3	27	19.8	24
		40		16	14	18.2	3	3	3	24	21.9	23
20	3	20	40	20	20	23.4	2	1	2	48	29	38
		30		20	18	23.2	3	3	3	47	33.5	40
		40		20	17	22.7	3	3	3	35	33.1	41

(1) – electromagnetic device (Figure 13a), (2) – electromagnetic device (Figure 13b), (3) – radar device (Figure 13c), * – values obtained from Equations (1) and (2).

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
