# Peer review of "Accuracy of Eddy-Current and Radar Methods Used in Reinforcement Detection"

_materials, 2019, doi:10.3390/ma12071168_

Reviewer 1 Report

As I understand, in this article the authors present numerous destructive and non-destructive tests “to describe possibilities of particular methods with emphasizing their advantages and limitations”. This version has improved from the previous one and has added a second stage of tests. Nevertheless, in my opinion, before this manuscript is ready for being published in Materials the authors should improve the following important issues:

a)      From my point of view the authors are not explaining clearly enough the novelty and importance of the article and research project in its field of expertise. And this is crucial. What’s the main contribution of this research project compared to previous projects? It should be more clearly presented in the abstract, introduction and conclusions so the potential readers can understand what contribution this research paper brings to this field of research.

b)     Furthermore, the abstract does not give a clearly enough summary of this research project main issues such as its objectives, parts, novelty and conclusions. The last sentence should be rewritten from my point of view.

c)      The first formulae and figures 1-3 are from previous research projects. Are all these figures along with all the related explanation needed? Because the article is becoming long and this part takes importance and room to the authors’ real contribution and novelty exposition and explanation.

d)     Section 5 title “Miejsca badań” is not correct, isn’t it?

e)      Conclusions should also improve. They should more clearly explain the main results of this research project and the novelty compared to previous research projects as said in point “a”. In this sense there is a worrying fact: the present improved version of this paper, which has a second stage of tests, has exactly the same conclusions compared to the previous version, which only had one stage of tests.

Some minor language spelling/typing should be reviewed like extra spacing in line 325.

Author Response

Below you can find our answers to the reviewer’s remarks in the order of their presence in the review.

a)       We corrected the abstract and the introduction. We paid attention to new elements and the originality of the article. We presented the purpose and scope of tests.

b)      The abstract has been completely changed.

c)       We removed the part with equations (1-3). However, Fig. 1 with remarks is left because they refer to few studies determining the accuracy of measurements taken with the electromagnetic method.

d)      The title of chapter 5 was translated.

e)      We modified conclusions and supplemented them.

We corrected captions to figures according to the reviewer’s suggestion.

Moreover, following suggestions of other reviewers:

1.  We additionally described new tests on 9 specimens that had been already tested. Tests were conducted on another side of models. Thus, we could consider how cover thickness affected test results.

2.  We corrected minor editorial mistakes.

Reviewer 2 Report

The manuscript presents a comparison of different methods for detecting reinforcement in concrete without destroying the samples. It conducts the tests on a lintel beam of autoclaved aerated concrete and on nine lightweight concrete specimens with two electromagnetic scanners and a ground-penetrating radar.

The paper provides some interesting conclusions. However, although tests were carried out in two different kinds of samples, the influence of the concrete cover could be analyzed. Thus, taking advantage of the nine samples, as the tests were carried out from the side with a concrete cover of 20 mm, the other side could be used to know if similar values are obtained with a different cover. This is the main point to be made to improve the paper: analyze the influence of the concrete cover. It must be taken into account that a cover of 20 mm is rejected in many European Standards. Please, explain also this last point (Reason for selecting such a narrow cover).

On the other hand, other minor points should be corrected.

Line 19-21. It mentions the devices that were employed: two electromagnetic scanners and a ground-penetrating radar. Then it is said that results from direct tests (which ones? The ones made with these devices?) were compared with results from non-destructive tests (NDT). So, it seems that results were compared with NDT. Later, on the introduction, it is explained that electromagnetic scanners and GPR are NDTs. Therefore, lines 19-21 can be confusing. It must be explained that selected devices are NDT.

Line 170. “The papers [28, 29] suggest…” This is not the way to indicate it. It should be “Wiwatrojanagul et al. [28] and Wiwatrojanagul et al. [29] suggest…”

Line 189. Instead of “presented in the papers [30,31,32], it should be indicated “…can be found elsewhere [30-32]”.

Line 227. Between “Lichtestein)” and “Profometer” a comma is needed.

Author Response

Following the reviewer’s suggestion, we additionally described new tests on 9 specimens that had been already tested. Tests were conducted on another side of models. Thus, we could consider how cover thickness affected test results. By analysing this remark, we found additional measurement errors for a small cover 10 mm, and quite a big cover (40 mm).

Responding to your question why we adopted the cover with thickness of 20 mm, we explain that table 4.4N in the standard Eurocode 2 recommends this thickness of cover for exposure classes X0, XC1- XC4 and XS1. So, such a cover can be applied in the majority of structures. The cover thickness 20 mm is very common in Poland. The article contains a relevant remark concerning this issue.

Other remarks of the reviewer on corrections in lines 170, 189, and 227 have been introduced.

Moreover, following suggestions of other reviewers:

1.       We modified the abstract, the introduction and the conclusions.

2.       We removed from the article the fragments containing equations that were not referred to in further parts of the article.

3.       We corrected minor editorial mistakes.

Reviewer 3 Report

I enjoyed reading this experimental work. The subject is of current relevance in civil and structural engineering and, hence, the paper falls within the scope of this journal.

Specific Recommendation for Revision:

1)      The innovative contribution of the study needs to be clarified in the abstract, and this reference:

a.  Zielińska, Monika, and Magdalena Rucka. "Non-Destructive Assessment of Masonry Pillars using Ultrasonic Tomography." Materials 11.12 (2018): 2543.

2)      Authors are invited to provide comments about fig 1 in order to explain why the error % is stabilized for concrete cover over 70mm;

3)      A figure reporting the test setup needs to be added;

4)      In fig. 12 the company that provided electromagnetic device is mentioned. Authors are invited to cover it;

5)      More and more often existing structures have previously been reinforced with FRP and FRCM type composite materials. In these cases, are the conclusions of this paper still valid? Please comment on these and add in the references. See:

a.   Ombres, L., & Verre, S. (2018). Shear performance of FRCM strengthened RC beams. Paper presented at the American Concrete Institute, ACI Special Publication, 2017-March (SP 324).

b.   Micelli, F., Cascardi, A., & Marsano, M. (2016, June). Seismic strengthening of a theatre masonry building by using active FRP wires. In Brick and Block Masonry: Proceedings of the 16th International Brick and Block Masonry Conference (pp. 753-761).

Author Response

Below you can find our answers to the reviewer’s remarks in the order of their presence in the review.

1)      We again wrote the abstract and the introduction. We paid attention to new elements and the originality of the article. We presented the purpose and scope of tests.

2)      Fig. 1 presents the measurement error for the rebar diameter. For the cover of 71 mm thickness, the error was nearly 100%. A relevant remark is in the article.

3)     The studies consisted in passing a transducer of the given instrument over the area of tested elements along already routed lines. Both area and line scans were taken. Figs 11 and 12 show scanned areas. We decided not to present the test stand because the application of three different instruments (different scanned areas and potentials of line or area scans) and two types of tested elements would require more than two images of test stands. Thus, the information could become unintelligible. But we added a photo (Fig. 10) presenting the transducer of one instrument on the tested lintel beam. Transducers used for the second type of tested elements are illustrated in Fig. 13.

4)      Company names of instruments have been rubbed out from all photos.

5)     The non-metallic reinforcement in the form of composite grid cannot be detected with the electromagnetic instruments as no electromagnetic induction occurs. The relevant remark and references to the literature are in the article. However, the non-metallic reinforcement can be detected by GPR scanners or ultrasonic tomography. We have already conducted such tests and the obtained results are very interesting. We tested concrete slabs reinforced with GFRP bar grid and masonry walls with non-metallic reinforcement in the form of fibreglass and basalt meshed laid in bed joints. We are going to write another article to the extra edition of Materials entitled “Non-Destructive Testing of Structures” (https://www.mdpi.com/journal/materials/special_issues/structures_NDT )

Moreover, following suggestions of other reviewers:

1.       We additionally described new tests on 9 specimens that had been already tested. Tests were conducted on another side of models. Thus, we could consider how cover thickness affected test results.

2.       We removed from the article the fragments containing equations that were not referred to in further parts of the article.

3.       We corrected minor editorial mistakes.

Round  2

Reviewer 1 Report

This version has improved a lot from the previous one and in my opinion this manuscript is ready for being published in Materials. Some language spelling/typing should be reviewed starting with the abstract, for example: “The aim of studies was to demonstrate”.

Author Response

Thank you for indicating the necessary corrections. We took into account all suggestions.

Reviewer 2 Report

Authors have included an analysis with variable concrete cover, as suggested, which really improves the quality of the conclusions. Moreover, following suggestions from other reviewers, the article can be published.

Some minor corrections:

Line 55. "Many reference papers present advantages and possibilities..." This sentence should be rewritten, something like: "Many papers in the literature..." Moreover, some examples of these papers must be included.

Line 95. The expression "The paper [11] describes tests..." is not adequate. According to the reference, it could be: "Drobiec et al. [11] describe tests..."

Line 97-98. Similarly, the expression "The paper [14] presented the analysis..." is not adequate. It could be "Sivasubramanian et al. [14] presented..."

Author Response

(The authors gave the same response as above.)
